# Gait Pattern in Charcot-Marie-Tooth Disease Type 1A According to Disease Severity

**DOI:** 10.3390/jpm13101473

**Published:** 2023-10-08

**Authors:** Jihyun Park, So Young Joo, Byung-Ok Choi, Dae-Hyun Kim, Jong Bum Park, Jong Weon Lee, Deog Young Kim

**Affiliations:** 1Department of Rehabilitation Medicine, Hallym University Dongtan Sacred Heart Hospital, College of Medicine, Hallym University, Hwaseong 18450, Republic of Korea; jhpark3620@gmail.com; 2Department of Rehabilitation Medicine, Hangang Sacred Heart Hospital, College of Medicine, Hallym University, Seoul 07247, Republic of Korea; anyany98@hallym.or.kr; 3Department of Neurology, Samsung Medical Center, School of Medicine, Sungkyunkwan University, Seoul 06351, Republic of Korea; bochoi@skku.edu; 4Department of Physical and Rehabilitation Medicine, Center for Prevention and Rehabilitation, Heart Vascular Stroke Institute, Samsung Medical Center, School of Medicine, Sungkyunkwan University, Seoul 06351, Republic of Korea; hohoho7490@gmail.com; 5Department of Rehabilitation Medicine, College of Medicine, Konyang University, Daejeon 35365, Republic of Korea; jbocean@konyang.ac.kr; 6Department of Rehabilitation Medicine, College of Medicine, Yonsei University, Seoul 03722, Republic of Korea; bickpjl@naver.com; 7Research Institute of Rehabilitation Medicine, College of Medicine, Yonsei University, Seoul 03722, Republic of Korea

**Keywords:** gait analysis, Charcot-Marie-Tooth disease, hereditary sensory motor neuropathy

## Abstract

The aim of this study was to evaluate the characteristics of gait patterns in Charcot-Marie-Tooth disease type 1A (CMT1A) patients according to disease severity. Twenty-two CMT1A patients were enrolled and classified into two groups, according to the disease severity. The healthy control group consisted of 22 subjects with no gait impairment. Full barefoot three-dimensional gait analysis with temporospatial, kinematic, and kinetic data was performed among the mild and moderate CMT1A group and the control group. Minimal hip abduction, maximal hip extension generation, peak knee flexion moment at stance, ankle dorsiflexion at initial contact, maximal ankle plantarflexion at push-off and maximal ankle rotation moment at stance in the CMT1A group showed a significant difference compared to the control group (*p* < 0.05). In the moderate group, there were greater maximal hip flexion angles in swing, and smaller dorsiflexion angles at initial contact compared to the control group and mild group. CMT patients had typical gait characteristics and their gait patterns were different according to severity. The analysis of gait patterns in patients with CMT1A will help to understand gait function and provide important information for the treatment of patients with CMT in the future.

## 1. Introduction

Charcot-Marie-Tooth disease (CMT), also known as progressive peroneal muscular atrophy or hereditary motor and sensory neuropathy (HMSN), was initially reported in 1886 by Charcot and Marie (France), and Tooth (UK). The disease is characterized by peroneal muscle atrophy as the primary manifestation. Since the initial description, it has been studied by researchers to understand its etiology and pathophysiology [1,2,3,4,5]. However, the exact cause and mechanisms of the disease remain elusive [6].

Charcot-Marie-Tooth disease is primarily an autosomal dominant, progressive, symmetric disorder of motor and sensory nerves, which predominantly manifests in the lower extremities. It is classified as a disorder of peripheral nerve, and the clinical presentation includes the symptoms of gradual muscular atrophy and sensory deficits, starting from the distal portions of the feet and hand and progressing towards the proximal extremities. The severity of the symptoms is assumed to correlate with the degree of axonal degeneration [7,8]. It is classified into seven categories based on manifestations, genetic inheritance patterns, electrophysiology, or results of pathologic changes from nerve biopsy [9]. Over the past few decades, with the discovery of various genes contributing to variability, molecular genetic examination has become an important diagnostic method for CMT disease. CMT Type 1 is demyelinating, and autosomal dominant phenotype currently subdivided into A, B, C, D, E, F, G, and HNPP based on the specific gene mutation and related phenotypes.

The most common subtype, constituting about 70% of cases of this disorder, is CMT1A. Its initial clinical presentation involves pain, muscle weakness, and foot deformities [10], followed by progression of weakness to affect leg muscles [11]. Such clinical progression leads to abnormal gait patterns and, in some cases, even results in the loss of ambulation. However, it is widely acknowledged that in the majority of cases, the clinical course is known to advance slowly. Krajewski et al. reported that out of forty-two walkable CMT patients, only three required the use of a wheelchair in the later time [7]. However, Pfeiffer et al., when analyzing the walking patterns of fifty patients, found that a mere five individuals exhibited a normal gait pattern [12].

Previous research studies on gait impairment and patterns in the context of CMT disease have predominantly focused on describing gait patterns within patient groups based on the individual presentation or genetic traits without objective measurements [13,14]. Additionally, some studies only involved a comparison of gait patterns between a control group and the CMT population [15,16]. Despite the significance of gait abnormalities in CMT disease, currently, there is a lack of studies analyzing differences in gait abnormalities among patient subgroups based on the disease severity. One study described the abnormal gait patterns among CMT disease subgroups, however, the subjects of the study were limited to children [17].

The objective of this study was to investigate the characteristics of gait patterns among a cohort of 22 individuals diagnosed with CMT1A. This investigation was conducted through the application of motion analysis, encompassing the measurement of kinematic and kinetic parameters. Additionally, a comparative analysis was performed between the gait patterns of CMT1A patients and those of the control group of individuals without CMT, to identify distinct gait features.

## 2. Materials and Methods

### 2.1. Patient Selection

The study included a cohort of 22 individuals diagnosed with CMT1A disease through physical examinations, family history assessments, electromyographic studies, and molecular genetic analyses. All participants in this study demonstrated the ability for independent ambulation. Individuals who were unwilling to participate in the study or those who experienced difficulty to maintain an upright posture, affecting their gait ability, were excluded.

### 2.2. Assessment of Disease Severity

Each participant underwent physical examinations, encompassing the observation of sensory and motor symptoms, along with assessments of limb sensory perception and muscle strength. Motor and sensory conduction studies, involving the measurement of compound muscle action potential and sensory action potential studies of tibial nerves, were conducted to calculate the CMT1A neuropathy score (CMTNS). The 22 participants were categorized into two groups based on the CMTNS, utilizing the criteria established by Shy [18]. Among them, 14 individuals with a score of 10 or below were classified into the mild group, while 8 individuals with a score of 11 or above were classified into the moderate group. The CMTNS is a scale developed to measure the natural clinical progression and treatment effectiveness of CMT disease, based on the total CMT neuropathy score, which was originally used for diabetic neuropathy and toxic neuropathy [19]. The score more effectively reflects the severity of CMT disease, and its reliability and validity have been substantiated, leading to its widespread clinical adoption.

### 2.3. Assessment of Gait Analysis

The temporospatial, kinetic, and kinematic data were obtained using a three-dimensional gait analyzer, VICON 370 (Oxford Metrics Ltd., Oxford, UK) with dual force-plate system (AMTI Inc., Watertown, MA, USA). According to the VICON Protocol, a set of 13 markers were attached during upright stationary posture, and participants were instructed to walk without assistance and without wearing shoes. The markers were bilaterally positioned at the following anatomical landmarks: between the second and third metatarsal heads, the lateral malleolus, the midpoint of the lateral aspect of fibula, the lateral epicondyle of femur (vertically below the greater trochanter), the midpoint of the lateral aspect of femur, the anterior superior iliac spine, and the first sacral vertebra. After inducing natural walking through multiple laps of an 8-meter track, participants were encouraged to walk a minimum of five times, ensuring a variety of walking cycles. All kinematic and kinetic parameters were acquired using Polygon software (Oxford Metric Ltd., Oxford, UK), which also calculated joint moments normalized to body mass. The temporospatial parameters were normalized using leg length [20]. To mitigate potential errors from infrared cameras, adjustments were made before conducting the assessments.

The characteristics of gait patterns in patients with CMT1A, divided into mild and moderate groups, were analyzed by comparing their temporospatial, kinematic, and kinetic parameters with those of the healthy control group.

### 2.4. Statistical Analysis

Statistical analyses were performed using the SPSS 25.0 statistical package (IBM Corp., Armonk, NY, USA). A Mann–Whitney test was used to evaluate the significance of differences for the characteristics among the groups. The analysis of variance (ANOVA) and post hoc Tukey’s test were used to analyze the differences of the general characteristics or parameters among the three groups. The significant difference was defined by *p* < 0.05.

## 3. Results

Among all the participants, the CMT1A disease group consisted of 22 individuals (fourteen males and eight females), with an average age of 29.6 years. There were no statistically significant differences in age or gender between the mild and moderate subgroups within the CMT1A disease group. The average CMTNS for the mild group was 5.9 points, while that of the moderate group was 17.5 points. The functional disability score [21] was 0.9 points and 3.0 points for the mild and moderate groups, respectively. Both the CMTNS and the functional disability score were significantly higher in the moderate group compared to the mild group (*p* < 0.05) (Table 1). Overall, 22 healthy adults that were gender- and age-matched were recruited as the control groups. There were no statistically significant differences in age or gender between the CMT1A disease group and the control group.

### 3.1. Characteristic Gait Patterns of CMT1A Disease

#### 3.1.1. Temporospatial Gait Parameters

The gait speed significantly decreased in the CMT1A disease group, with a mean of 1.05 m/s, compared to a mean of 1.14 m/s in the control group (*p* < 0.05). The step length was significantly shorter in the CMT1A disease group, 0.56 m, than in the control group, 0.61 m, (*p* < 0.05). However, there were no significant differences between the two groups in terms of cadence and step time (Table 2).

#### 3.1.2. Hip Kinematics and Kinetics

The maximal hip adduction angle was significantly smaller in the CMT1A disease group than in the control group. The hip power generation in stance was significantly higher in the CMT1A disease group than in the control group. The hip power generation at pre-swing was significantly lower in the CMT1A disease group than in the control group. Other hip kinematics and kinetics showed no significant differences between the two groups (Table 3, Figure 1).

#### 3.1.3. Knee Kinematics and Kinetics

The minimal knee flexion angle in stance was significantly smaller in the CMT1A group than in the control group. The maximal knee varus angle in stance was significantly greater in the CMT1A group than in the control group. The maximal knee flexion moment in stance was significantly lower in the CMT1A group than in the control group. The maximal knee power generation in stance was significantly lower in the CMT1A disease group than in the control group. The maximal knee abduction moment in stance was significantly higher in the CMT1A disease group than in the control group. There were no significant differences in the other knee kinematics and kinetics between the two groups (Table 4, Figure 2).

#### 3.1.4. Ankle Kinematics and Kinetics

The ankle dorsiflexion angle at initial contact was significantly increased in the CMT1A group than in the control group. The maximal ankle dorsiflexion angle in stance was significantly smaller in the CMT1A group than in the control group. At the 98% point of the gait cycle, the dorsiflexion angle was significantly decreased in the CMT1A group than in the control group. The maximal ankle plantarflexion angle at push-off was significantly smaller in the CMT1A group than in the control group, indicating a reduction in plantarflexion in CMT1A patients.

No significant differences were observed between the control group and the CMT1A disease group in terms of the peak ankle dorsiflexion moment and the maximal ankle eversion (abduction) moment. However, the peak ankle plantarflexion moment was significantly lower in the CMT1A disease group than in the control group. The maximal ankle inversion (equals to minimal ankle abduction) moment was significantly higher in the CMT1A disease group than in the control group. The maximal ankle rotation (internal rotation) moment in stance was significantly higher in the CMT1A disease group than in the control group. The maximal ankle power generation was significantly lower, and the maximal ankle power absorption was significantly higher in the CMT1A disease group than in the control group. There were no significant differences in the other ankle kinematics and kinetics between the two groups (Table 5, Figure 3).

### 3.2. Gait Patterns by CMT1A Disease Severity

#### 3.2.1. Temporospatial Gait Parameters

The gait speed in the moderate group was significantly slower, with a mean of 0.95 m/s, compared to the control group and the mild group. The gait speed in the mild group was slower, compared to the control group; however, the difference was not statistically significant. Cadence was significantly larger in the mild group than in the control group, while the moderate group showed a significantly smaller cadence than those of the control group and the mild group. Step length was significantly decreased in both the mild group and the moderate group than in the control group, while there was no statistically significant difference in step length between the mild and the moderate groups. The step time significantly increased in the moderate group, compared to both the control and mild groups (Table 2).

#### 3.2.2. Hip Kinematics and Kinetics

The maximal hip flexion angle in the swing was significantly increased in the moderate group than in the control group and the mild group. The maximal hip external rotation angle was significantly greater in the moderate group than in the mild group. There were no significant differences in the minimal hip flexion angle (maximal hip extension angle) and the maximal hip external rotation angle between the control group and each CMT1A group. Among the hip kinetic parameters, there were no significant differences between the control group and each CMT1A disease group for the maximal hip flexion and extension moment in stance, the maximal hip power generation in stance and at pre-swing, the maximal hip adduction and abduction moment in stance, and the hip rotation moment in stance (Table 3, Figure 1).

#### 3.2.3. Knee Kinematics and Kinetics

The knee flexion angle at the initial contact was significantly greater in the moderate group than in the control group and the mild group. The peak knee flexion angle in swing was significantly greater in the moderate group than in the control group and the mild group. The maximal knee varus angle in the swing was significantly increased in the mild group and the moderate group compared to the control group. However, there was no significant difference in the maximal knee varus angle in swing between the CMT1A disease groups. There were no significant differences between the CMT1A disease groups in terms of the maximal knee flexion angle at midstance, the maximal knee varus and valgus angle in stance, and the maximal knee varus and valgus in the swing. The peak knee flexion moment in stance was significantly lower in the mild group and the moderate group compared to the control group. However, no significant differences were observed between the CMT1A groups. The maximal knee abduction moment in stance was significantly increased in both the mild group and the moderate group than in the control group with the increase being more significant in the mild group than in the moderate group. There were no statistically significant differences in the maximal knee extension moment and the maximal knee power generation in stance between the CMT1A disease groups (Table 4, Figure 2).

#### 3.2.4. Ankle Kinematics and Kinetics

The maximal ankle dorsiflexion angle in stance was significantly smaller in the mild group than in the control group, while it was significantly increased in the moderate group compared to the mild group. The ankle dorsiflexion angle at initial contact was significantly smaller in both the mild group and the moderate group than in the control group. Additionally, the ankle dorsiflexion angle at the initial contact was significantly smaller in the moderate group compared to the mild group. At the 98% point of the gait cycle, the ankle dorsiflexion angle was significantly decreased in both the mild group and the moderate group than in the control group. The maximal ankle plantarflexion angle at push-off was significantly smaller in both the mild group and the moderate group compared to the control group, indicating a decrease in ankle plantarflexion, while no significant differences were observed between the mild and moderate groups.

The peak ankle dorsiflexion moment in stance was significantly lower in the moderate group, compared to both the control group and the mild group. The peak ankle plantarflexion moment in stance was significantly lower in both the mild group and the moderate group, compared to the control group, while there were no significant differences between the mild and moderate groups. There were no significant differences in the maximal ankle inversion (the minimal ankle abduction) moment between the control group and each CMT1A disease group. The maximal ankle eversion (abduction) moment in stance was significantly higher in the mild group than in the control group. The maximal ankle internal rotation moment (the maximal ankle rotation moment) in stance was significantly higher in both the mild group and the moderate group than in the control group, with no significant differences observed between the CMT1A disease groups (Table 5, Figure 3).

## 4. Discussion

In this study, through three-dimensional gait analysis, a comprehensive comparison of temporospatial, kinematic, and kinetic characteristics of gait differences between the control group and the CMT1A disease group, and between the CMT1A disease subgroups based on disease severity was conducted. Gait speed exhibited a significant decrease in the CMT1A disease group compared to the control group, which can be attributed to the deviations in the kinematic and kinetics of gait patterns. Specifically, the reduced ankle plantarflexor force at push-off in CMT patients is believed to contribute to this discrepancy [16]. On the contrary, the slower gait speed could potentially influence the kinematic and kinetic changes in walking. However, the observed deviations in the gait pattern of the CMT1A disease group in this study were not consistent with the alterations seen in walking due to a slower gait speed in the control group [22]. Instead, these deviations more closely resembled the gait deviations resulting from weakened hip and ankle muscle powers, rather than those occurring with a slower gait pattern in healthy individuals [23]. Furthermore, a decrease in various ankle joint angles was observed in the CMT1A disease group, including the maximal ankle dorsiflexion angle, the dorsiflexion angle at 98% of the gait cycle, the ankle dorsiflexion at initial contact, and the maximal ankle plantarflexion at push-off with an increase in the maximal ankle rotation in stance. These findings explain the occurrence of foot drop due to the impairment of ankle dorsiflexor muscles. Consequently, this phenomenon leads to an increased ankle plantarflexion angle at initial contact. The compromised function of the gastrocnemius and soleus muscles result in reduced ankle plantarflexion at push-off and diminished dorsiflexion power during the late stance phase. This inadequate ankle dorsiflexion, along with insufficient ankle plantarflexion, resembles the pattern observed by Vinci and Perelli [11] in CMT patients, where there was nerve degeneration progresses from distal to proximal within the lower extremities, leading to symptomatic changes in ankle kinematics due to muscle weakness and atrophy. They attributed these alterations to an imbalanced function of the ankle evertor muscles, causing a supination deformity of the ankle joint. This explanation aligns with the findings of this study, which showed increased ankle internal rotation moment [24,25].

This study identified a reduction in knee flexion angle in stance and an increase in knee extension power generation in stance. Kendall and McCreary et al. [26] suggested that the biomechanism of knee joint hyperextension correlates with muscular dysfunction caused by compromised gastrocnemius and soleus muscle. These findings are parallel to the results observed in this study, where the deviations in knee kinematics could be attributed specifically to the weakened ankle plantarflexor muscles. The previous researchers expounded on compensatory mechanisms triggered by muscle imbalances around the ankle joint, leading to foot supination [26,27]. This compensation induces internal rotation of the shank relative to the thigh, aiding in enhancing foot progression during the loading response phase. The mechanism involving hip external rotation during shank progression, contributing to improved foot progression in foot supination position was also elucidated [17]. In normal gait, the hip joint maintains an adducted position as the center of mass shifts over the weight-bearing lower limb in the stance phase [22]. A decrease in hip adduction signifies a wider base of support in walking, a phenomenon observed in patients with neurological deficits like CMT or individuals with ankle instability [28,29]. This phenomenon is consistent with the findings in this study, where a decrease in the maximal hip adduction in CMT1A disease group aligns with the altered hip kinematics in the previous studies [17,30].

In this study, gait speed was significantly slower in the moderate group compared to the mild group, and in the swing phase, the maximal hip flexion angle was greater in the moderate group. Additionally, distinct characteristics were noted in the moderate group, including increased knee flexion angle at initial contact, the maximal knee flexion angle in swing, decreased ankle dorsiflexion angle at the 98% point of the gait cycle and initial contact, and a reduced peak ankle dorsiflexion moment in stance, all compared to the mild group. These gait patterns demonstrate that as the disease progresses, the severity of foot drop becomes more profound [31,32]. In cases where there is mild muscle weakening of the ankle dorsiflexors, particularly at initial contact, impaired eccentric contractions of the ankle dorsiflexor muscles can lead to foot slap. Moreover, in instances of more pronounced muscle weakness, gait imbalance with instability manifests at midstance, resulting in increased pronation and supination of the foot, coupled with dragging of the foot in the swing phase. This phenomenon is associated with a severe hip flexion and ankle plantarflexion, called steppage gait, which was also observed in this study and is explained by the progressive nature of the disease condition with the extent of food drop [17,30,33,34].

This study has several limitations. First, the cross-sectional design nature of this study may pose challenges in accurately assessing the changes in gait patterns over time. While this study investigated gait patterns and variations based on the severity of CMT1A disease in a cohort, it did not involve a prospective, longitudinal approach to track serial progression in gait abnormalities over an extended period individually. Therefore, the ability to precisely infer alteration linked to disease progression might be constrained. Second, the number of patients was relatively small even though we tried to recruit patients through a special clinic with the largest CMT cohort. This could be attributed to the fact that CMT is a rare disease. In forthcoming studies, it is anticipated that these considerations will be addressed, and a large-scale prospective, longitudinal study will be conducted for a more comprehensive analysis of the progressive nature of gait characteristics.

## 5. Conclusions

This study shows that patients with CMT1A present with a unique gait pattern that correlate with the severity of the disease. Alterations of gait patterns in the proximal part of the lower limb were correlated to the disease severity. Based on the findings of this study, it was posited that comprehending the distinctive gait patterns linked to different levels of disease severity could offer valuable insights into addressing joint deformities and musculoskeletal discomfort in lower limbs. Consequently, this could enhance therapeutic interventions.

## Figures and Tables

**Figure 1 jpm-13-01473-f001:**
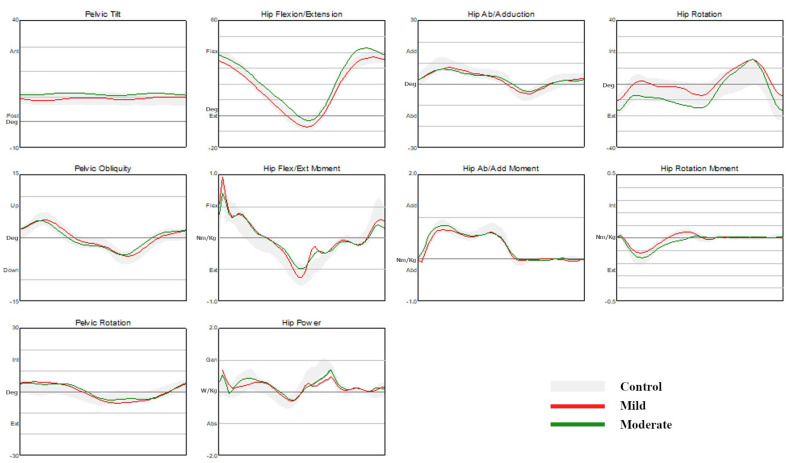
Comparison of kinematic and kinetic data of the pelvis and the hip joint.

**Figure 2 jpm-13-01473-f002:**
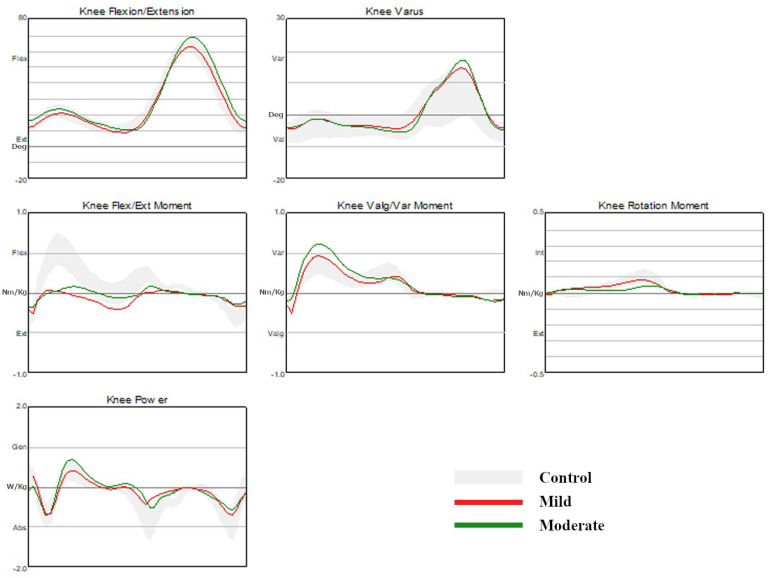
Comparison of kinematic and kinetic data of the knee joint.

**Figure 3 jpm-13-01473-f003:**
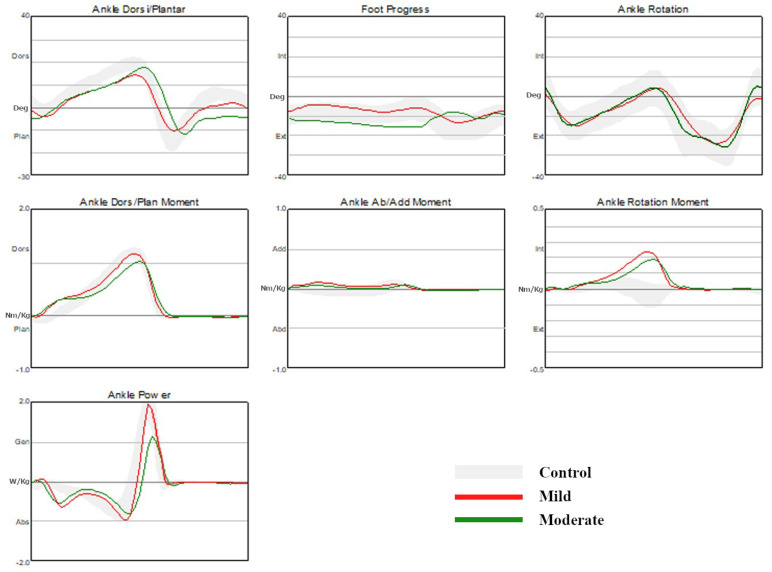
Comparison of kinematic and kinetic data of the ankle joint.

**Table 1 jpm-13-01473-t001:** Characteristics of the subjects.

	Control Group	CMT1A Group		
		**CMT1A Total**	**Mild Group**	**Moderate Group**
Age (years)	21.3 ± 1.6	29.6 ± 17.0	22.1 ± 12.0	42.8 ± 16.5 *^†^
Sex (female:male)	14:8	14:8	10:4	4:4
Neuropathy score			5.93 ± 2.4	17.5 ± 6.6 ^†^
Functional disability score			0.9 ± 0.5	3.0 ± 1.4 ^†^

Values are expressed as mean ± standard deviation. *p*-values were calculated by Mann–Whitney-U test. * *p*-value < 0.05 vs. control group; ^†^ *p*-value < 0.05 vs. mild group.

**Table 2 jpm-13-01473-t002:** Comparison of temporospatial parameters between the CMT1A patient subgroups and control group.

	Control Group	CMT1A Group		
		**CMT1A Total**	**Mild Group**	**Moderate Group**
Gait speed (m/s)	1.14 ± 0.10	1.05 ± 0.14 *	1.11 ± 0.11	0.95 ± 0.12 *^†^
Cadence (steps/min)	110.50 ± 5.84	111.38 ± 9.58	115.23 ± 7.63 *	104.63 ± 9.05 *^†^
Step length (m)	0.61 ± 0.05	0.56 ± 0.07 *	0.58 ± 0.07 *	0.54 ± 0.07 *
Step time (s)	0.54 ± 0.03	0.54 ± 0.05	0.52 ± 0.04	0.58 ± 0.06 *^†^

Values are expressed as mean ± standard deviation. *p*-values were calculated by ANOVA with post-hoc Tukey’s-b test. * *p*-value < 0.05 vs. control group; ^†^ *p*-value < 0.05 vs. mild group.

**Table 3 jpm-13-01473-t003:** Comparison of kinematic and kinetic data of the hip joint between CMT1A patient subgroups and control group.

	Control Group	CMT1A Group		
		**CMT1A Total**	**Mild Group**	**Moderate Group**
Maximal hip flexion in swing (°)	37.81 ± 3.57	40.22 ± 7.70	37.55 ± 5.43	44.89 ± 8.98 *^†^
Minimal hip flexion in stance (°)	−4.86 ± 4.34	−5.84 ± 6.57	−7.35 ± 6.09	−3.20 ± 6.72
Hip flexion at initial contact (°)	37.32 ± 3.71	34.63 ± 10.07	32.34 ± 11.46 *	38.63 ± 5.25 ^†^
Maximal Hip abduction (°)	−5.21 ± 2.67	−4.63 ± 4.23	−4.82 ± 3.70	−4.30 ± 5.15
Minimal hip abduction (°)	10.14 ± 3.18	8.23 ± 3.56 *	8.37 ± 3.62	8.00 ± 3.56
Maximal hip external rotation (°)	16.63 ± 7.28	16.17 ± 10.00	13.00 ± 7.04	21.73 ± 12.07 ^†^
Maximal hip flexion moment in stance (Nm/kg)	0.94 ± 0.36	0.95 ± 0.29	1.0 ± 0.28	0.87 ± 0.29
Maximal hip extension moment in stance (Nm/kg)	−0.65 ± 0.13	−0.60 ± 0.21	−0.61 ± 0.23	−0.57 ± 0.18
Maximal hip flexion generation in stance (Watt/kg)	1.01 ± 0.33	1.60 ± 0.64 *	1.76 ± 0.62	1.32 ± 0.60
Maximal hip extension generation at pre-swing (Watt/kg)	0.35 ± 0.26	0.13 ± 0.24 *	0.09 ± 0.18	0.21 ± 0.30
Maximal hip adduction moment in stance (Nm/kg)	0.86 ± 0.13	0.85 ± 0.25	0.80 ± 0.20	0.93 ± 0.32
Minimal hip adduction moment in stance (Nm/kg)	0.04 ± 0.04	0.04 ± 0.05	0.04 ± 0.05	0.03 ± 0.04
Maximal hip rotation moment in stance (Nm/kg)	0.07 ± 0.02	0.08 ± 0.04	0.08 ± 0.05	0.07 ± 0.04
Minimal hip rotation moment in stance (Nm/kg)	−0.17 ± 0.05	−0.15 ± 0.10	−0.13 ± 0.07	−0.18 ± 0.13

Values are expressed as mean ± standard deviation. *p*-values were calculated by ANOVA with post hoc Tukey’s-b test. * *p*-value < 0.05 vs. control group; ^†^ *p*-value < 0.05 vs. mild group.

**Table 4 jpm-13-01473-t004:** Comparison of kinematic and kinetic data of the knee joint between CMT1A patient subgroups and control group.

	Control Group	CMT1A Group		
		**CMT1A Total**	**Mild Group**	**Moderate Group**
Knee flexion at initial contact (°)	12.79 ± 3.80	13.88 ± 7.44	12.04 ± 4.93	17.11 ± 9.87 *^†^
Peak knee flexion at midstance (°)	23.18 ± 4.67	21.65 ± 11.99	19.75 ± 10.01	24.97 ± 14.59
Knee flexion at toe-off (°)	42.18 ± 6.91	41.35 ± 8.84	38.10 ± 5.42	47.05 ±10.79 ^†^
Peak knee flexion in swing (°)	65.56 ± 3.80	66.35 ± 6.76	63.97 ± 5.01	70.52 ± 7.53 *^†^
Minimal knee flexion in stance (°)	10.41 ± 2.81	7.71 ± 7.98 *	6.85 ± 6.21 *	9.21 ± 10.45
Maximal knee varus in stance (°)	2.25 ± 5.55	4.15 ± 8.97	3.66±8.68	5.01 ± 9.69
Minimal knee varus in stance (°)	−5.82 ± 3.71	−5.03 ± 6.89	−3.93 ± 8.04	−6.96 ± 3.71
Maximal knee valgus in swing (°)	−5.58 ± 3.24	−4.92 ± 3.21	−4.36 ± 2.94	−5.91 ± 3.53
Maximal knee varus in swing (°)	9.73 ± 7.23	17.09 ± 8.95 *	16.63 ± 8.10 *	17.88 ± 10.53 *
Peak knee flexion moment in stance (Nm/kg)	0.58 ± 0.21	0.20 ± 0.18 *	0.19 ± 0.18 *	0.22 ± 0.18 *
Peak knee extension moment in stance (Nm/kg)	−0.35 ± 0.16	−0.34 ± 0.18	−0.34 ± 0.13	−0.33 ± 0.26
Maximal knee abduction moment in stance (Nm/kg)	−0.07 ± 0.05	−0.23 ± 0.14 *	−0.27 ± 0.14 *	−0.15 ± 0.10 *^†^

Values are expressed as mean ± standard deviation. *p*-values were calculated by ANOVA with post hoc Tukey’s-b test. * *p*-value < 0.05 vs. control group; ^†^ *p*-value < 0.05 vs. mild group.

**Table 5 jpm-13-01473-t005:** Comparison of kinematic and kinetic data of the ankle joint between CMT1A patient subgroups and control group.

	Control Group	CMT1A Group		
		**CMT1A Total**	**Mild Group**	**Moderate Group**
Maximal ankle dorsiflexion in stance (°)	20.15 ± 3.34	16.87 ± 5.02 *	15.31 ± 4.12 *	19.60 ± 5.42 ^†^
Ankle dorsiflexion at 98% gait cycle (°)	4.47 ± 3.09	−0.64 ± 5.27 *	1.22 ± 3.19 *	−3.90 ± 6.59 *^†^
Ankle dorsiflexion at initial contact (°)	3.36 ± 2.71	−2.35 ± 5.16 *	−1.00 ± 3.48 *	−4.73 ± 6.72 *^†^
Maximal ankle plantarflexion at push-off (°)	10.31 ± 6.28	5.19 ± 6.53 *	5.68 ± 6.12 *	4.33 ± 7.32 *
Peak ankle dorsiflexion moment in stance (Nm/kg)	1.24 ± 0.10	1.15 ± 0.28	1.22 ± 0.22	1.02 ± 0.32 *^†^
Peak ankle plantarflexion moment in stance (Nm/kg)	−0.12 ± 0.05	−0.05 ± 0.05 *	−0.05 ± 0.05 *	−0.04 ± 0.04 *
Maximal ankle abduction moment in stance (Nm/kg)	−0.05 ± 0.04	−0.05 ± 0.07	−0.05 ± 0.06	−0.07 ± 0.08
Minimal ankle abduction moment in stance (Nm/kg)	0.08 ± 0.06	0.13 ± 0.12 *	0.14 ± 0.13 *	0.12 ± 0.12
Maximal ankle rotation moment in stance (Nm/kg)	0.05 ± 0.04	0.24 ± 0.15 *	0.26 ± 0.17 *	0.22 ± 0.10 *
Minimal ankle rotation moment in stance (Nm/kg)	−0.06 ± 0.06	−0.03 ± 0.04 *	−0.03 ± 0.03 *	−0.04 ± 0.05
Maximal ankle power generation (Watts/kg)	2.64 ± 0.48	1.80 ± 0.73 *	2.05 ± 0.53 *	1.36 ± 0.85 *^†^
Maximal ankle power absorption (Watts/kg)	−0.84 ± 0.21	−1.03 ± 0.32 *	−1.06 ± 0.33 *	−0.99 ± 0.32

Values are expressed as mean ± standard deviation. *p*-values were calculated by ANOVA with post hoc Tukey’s-b test. * *p*-value < 0.05 vs. control group; ^†^ *p*-value < 0.05 vs. mild group.

## Data Availability

The data presented in this study are available on request from the corresponding author under the condition of confidentiality. The data are not publicly available due to privacy or ethical restrictions.

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
