# Peer review of "Gait Pattern in Charcot-Marie-Tooth Disease Type 1A According to Disease Severity"

_jpm, 2023, doi:10.3390/jpm13101473_

Round 1

Reviewer 1 Report

Overall this is an important and well conducted study evaluating gait in individuals with CMT1A. I only have a few comments for the authors to address:

1. Please confirm the age range of the CMT cohort? Control range was 20-60. Was CMT also that age range and therefore only adult patients or where any children included? I suggest including ranges in Table 1 for whole group as well as mild/moderate group as surprising no significant difference in age between groups given the means.

2. I suggest removing the angles from text as all listed in the tables so not necessary to repeat – just report results without the numbers.

3. Introductions states “Despite the significance of gait abnormalities in CMT disease, currently, there is a lack of studies analyzing differences in gait abnormalities among patient subgroups based on the disease severity.” But a reference you use later (Wojciechowski et al, 2017) does anlyze differences based on severity. Please address this in introduction or as a comparison in discussion.

4. Please expand on methods for gait analysis (for example – were force plates used to capture kinetic data as this is not mentioned; data processing specifics such as if temporal spatial measures were normalised to leg length and if not were heights/leg lengths similar between groups?, etc)

5. Reference not inserted correctly in the below sentence from discussion: This inadequate ankle dorsiflexion, along with insufficient ankle plantarflexion, resembles the pattern observed by Vinci and Perelli11 in CMT patients

N/A

Author Response

I would like to express my appreciation to the reviewers for their invaluable comments, which have greatly contributed to improving the manuscript in a more informative direction.

Review 1>

  1. Please confirm the age range of the CMT cohort? Control range was 20-60. Was CMT also that age range and therefore only adult patients or where any children included? I suggest including ranges in Table 1 for whole group as well as mild/moderate group as surprising no significant difference in age between groups given the means.

>Thank you for your insightful comments. Your comment was correct that the age range for CMT ‘and’ control groups combined corresponds to the written range. The CMT group includes few children under 20. While there was no significant difference between mild CMT and control groups at the median, there was a significant difference in the age range when comparing the moderate CMT and control groups. There was a missing indication during statistical analysis, and upon your observation, we reanalyzed and corrected this oversight in the manuscript.

  1. I suggest removing the angles from text as all listed in the tables so not necessary to repeat – just report results without the numbers.

>I have revised the manuscript to focus more on the results themselves, removing the detailed angles and related information that can be found in the tables within the “Results” section.

  1. Introductions states “Despite the significance of gait abnormalities in CMT disease, currently, there is a lack of studies analyzing differences in gait abnormalities among patient subgroups based on the disease severity.” But a reference you use later (Wojciechowski et al, 2017) does anlyze differences based on severity. Please address this in introduction or as a comparison in discussion.

>As you mentioned, I have added this information to “Line 51-53” and described the described the issue with a focus on the differences. Thank you for your insightful comments that helped improve the content.

  1. Please expand on methods for gait analysis (for example – were force plates used to capture kinetic data as this is not mentioned; data processing specifics such as if temporal spatial measures were normalised to leg length and if not were heights/leg lengths similar between groups?, etc)

>I added the force-plate system on “Line 8”.

>I included an explanation about the normalization method on “Line 93-94”.

>The presented data are normalized based on each subject’s weight and leg length.

  1. Reference not inserted correctly in the below sentence from discussion: This inadequate ankle dorsiflexion, along with insufficient ankle plantarflexion, resembles the pattern observed by Vinci and Perelli11 in CMT patients

>I have corrected the mistakenly entered numbers on “Line 243”.

Reviewer 2 Report

Jihyun et al. revealed that CMT 1A patients had unique gait characteristics, which are correlated with the severity of the disease, This is an interesting study, however, there are still some minor problems:

1. Please replace CMT with CMT1A in the methods, results, and discussion sections.

2. If possible, please provided genetic analyses in the supplemental materials.

3. Moderate editing of English language required. For example:

Lines 109-102: The description may be more appropriate: 22 healthy adults with gender- and age- matched were recruited as the control groups.

Lines 304-305: The description may be more appropriate: This study shows that  patients with CMT 1A present with a unique gait pattern that correlate with the severity of the disease.

Moderate editing of English language required.

Author Response

I would like to express my appreciation to the reviewers for their invaluable comments, which have greatly contributed to improving the manuscript in a more informative direction.

Review 2>

  1. Please replace CMT with CMT1A in the methods, results, and discussion sections.

I have replaced all occurrences of ‘CMT’ specifying CMT1A with ‘CMT1A’ throughout the entire text. Thank you for the helpful suggestion.

  1. If possible, please provided genetic analyses in the supplemental materials.

I regret to inform that due to restricted access and limited permissions regarding the mutational analysis of specific regions for this study, we were unable to include this information in the open-source publication.

  1. Moderate editing of English language required. For example:

Lines 109-102: The description may be more appropriate: 22 healthy adults with gender- and age- matched were recruited as the control groups.

>I have corrected the sentence that you recommended on Line 113-114.

Lines 304-305: The description may be more appropriate: This study shows that  patients with CMT 1A present with a unique gait pattern that correlate with the severity of the disease.

 >I have made the necessary revision for the sentence on Line 296-297.

I extend my sincere gratitude to you for your thoughtful and appropriate recommendations.